# Characteristics of Classical Swine Fever Virus Variants Derived from Live Attenuated GPE^−^ Vaccine Seed

**DOI:** 10.3390/v13081672

**Published:** 2021-08-23

**Authors:** Taksoo Kim, Loc Tan Huynh, Shizuka Hirose, Manabu Igarashi, Takahiro Hiono, Norikazu Isoda, Yoshihiro Sakoda

**Affiliations:** 1Laboratory of Microbiology, Department of Disease Control, Faculty of Veterinary Medicine, Hokkaido University, Sapporo 060-0818, Hokkaido, Japan; ts-kim@vetmed.hokudai.ac.jp (T.K.); huynhtanloc@vetmed.hokudai.ac.jp (L.T.H.); shizuka-1612asteroid@eis.hokudai.ac.jp (S.H.); hiono@vetmed.hokudai.ac.jp (T.H.); nisoda@vetmed.hokudai.ac.jp (N.I.); 2Division of Global Epidemiology, International Institute for Zoonosis Control, Hokkaido University, Sapporo 001-0020, Hokkaido, Japan; igarashi@czc.hokudai.ac.jp; 3International Collaboration Unit, International Institute for Zoonosis Control, Hokkaido University, Sapporo 001-0020, Hokkaido, Japan

**Keywords:** classical swine fever virus, GPE^−^, live attenuated vaccine, vaccine seed, variant, NS5B, type I interferon

## Abstract

The GPE^−^ strain is a live attenuated vaccine for classical swine fever (CSF) developed in Japan. In the context of increasing attention for the differentiating infected from vaccinated animals (DIVA) concept, the achievement of CSF eradication with the GPE^−^ proposes it as a preferable backbone for a recombinant CSF marker vaccine. While its infectious cDNA clone, vGPE^−^, is well characterized, 10 amino acid substitutions were recognized in the genome, compared to the original GPE^−^ vaccine seed. To clarify the GPE^−^ seed availability, this study aimed to generate and characterize a clone possessing the identical amino acid sequence to the GPE^−^ seed. The attempt resulted in the loss of the infectious GPE^−^ seed clone production due to the impaired replication by an amino acid substitution in the viral polymerase NS5B. Accordingly, replication-competent GPE^−^ seed variant clones were produced. Although they were mostly restricted to propagate in the tonsils of pigs, similarly to vGPE^−^, their type I interferon-inducing capacity was significantly lower than that of vGPE^−^. Taken together, vGPE^−^ mainly retains ideal properties for the CSF vaccine, compared with the seed variants, and is probably useful in the development of a CSF marker vaccine.

## 1. Introduction

Classical swine fever (CSF) is a highly contagious disease of domestic pigs and wild boar caused by the classical swine fever virus (CSFV). CSFV is a member of the genus *Pestivirus* within the family *Flaviviridae*, with economically important animal pathogens, such as bovine viral diarrhea virus (BVDV) and border disease virus. CSFV carries a positive-sense single-stranded RNA genome of approximately 12.3 kb long with one large open reading frame (ORF) flanked by 5′ and 3′ untranslated regions (UTRs). The ORF encodes a single polyprotein of approximately 4000 amino acids. It is cleaved by cellular and viral proteases co- and post-translationally into 12 mature proteins (N^pro^, C, E^rns^, E1, E2, p7, NS2, NS3, NS4A, NS4B, NS5A, and NS5B) [1].

In Japan during the 20th century, the control and eradication of CSF advanced dramatically with the aid of a vaccine produced from live attenuated vaccine strain, GPE^−^. The GPE^−^ strain developed in Japan is based on the highly virulent strain, ALD, through 142 passages in swine testicle cells, 36 passages in bovine testicle cells, and 32 passages in the primary guinea pig kidney (pGPK) cells, combined with biological cloning [2]. By vaccinating domestic pigs with the GPE^−^ vaccine for about 30 years, Japan successfully obtained the World Organization for Animal Health (OIE) CSF-free status in 2007. However, the first CSF outbreak in 26 years occurred in Gifu Prefecture, Japan, in September 2018 [3,4]. Responding to the spread of CSF in surrounding areas, the vaccination program for domestic pigs using the GPE^−^ vaccine commenced in October 2019, and a systematic strategy to re-eradicate CSF is still being conducted.

Some marker vaccines, referred to as differentiating infected from vaccinated animals (DIVA) vaccine, for CSF, were developed as the leading-edge alternative to live vaccines [5,6]. The marker vaccine induces vaccine-specific antibodies only in vaccinated pigs. Combined with antibody diagnostics, such as enzyme-linked immunosorbent assays (ELISA), these vaccine-specific antibodies can be detected and differentiated from the antibodies against a field virus strain. Conversely, the absence of these specific antibodies indirectly helps identify pigs infected with a field strain. As this differentiation can be performed even under situations where the vaccination is practiced, the marker vaccine serves as a highly aspired countermeasure for CSF, especially in CSF-epidemic countries heading toward the disease-free status. In the case of Japan, a marker vaccine with the backbone of the safety and effectiveness-guaranteed GPE^−^ strain would surely boost the ongoing eradication campaign.

We have previously established the reverse genetics system of the GPE^−^ strain. Its infectious cDNA clone, vGPE^−^ helped elucidate CSF pathogenesis [7,8,9]. The attenuation of the GPE^−^ strain is represented by restricted propagation in pigs. It causes transient viremia in pigs and results in a slight virus propagation, exclusively in tonsils, which are the primary propagation site of CSFV [7,9]. Amino acids in N^pro^, E2, and NS4B reportedly explain the underlying mechanisms of attenuation. Particularly, its N^pro^ cannot prevent type I interferon (IFN) induction and confers the exaltation of Newcastle disease virus (END) phenomenon negative (END^–^) phenotype on itself, which is a distinct safety marker [7,10,11,12]. Despite these findings, vGPE^−^ was generated based on the nucleotide sequence of a reference GPE^−^ strain, which was cloned from the virus in a vaccine vial by limiting dilution after six passages in swine kidney-derived FS-L3 cells [13]. As a result, the vGPE^−^ strain possesses 10 amino acid substitutions, compared to the original GPE^−^ vaccine seed, called the master seed GPE^−^ [14]. Although it is now impossible to have possession of the intact master seed GPE^−^ for research use, the effects of these 10 amino acid substitutions on the virus characteristics need to be investigated to support the quality control of the GPE^−^ vaccine. Moreover, that investigation would also propose the present best vaccine platform for a new marker vaccine.

In the present study, we aimed to generate a clone that possesses the identical amino acid sequence to the master seed GPE^−^. Contrary to our expectations, the infectious virus was not rescued from the cDNA plasmid, carrying the full-length master seed GPE^−^ genome. This was attributed to the dysfunctionality of viral replication, which resulted from an amino acid substitution in NS5B. Accordingly, two replication-competent clone viruses were prepared as master seed GPE^−^ variants by re-substituting the corresponding amino acid. Their viral characteristics were evaluated both in vitro and in vivo, along with the vGPE^−^ strain.

## 2. Materials and Methods

### 2.1. Cells and Viruses

The SK-L cells were cultured in Eagle’s minimum essential medium (EMEM) (Nissui Pharmaceutical, Tokyo, Japan) supplemented with 0.295% tryptose phosphate broth (TPB) (Becton Dickinson, Franklin Lakes, NJ, USA), 10 mM N,N-bis-(2-hydroxyethyl)-2-aminoethanesulfonic acid (BES) (Sigma-Aldrich, Saint Louis, MO, USA), and 10% horse serum (HS) (Thermo Fisher Scientific, Waltham, MA, USA). The CPK cells [15] were cultured in EMEM supplemented with 0.295% TPB, 10 mM BES, and 5% HS. The SK6-MxLuc cells carrying a Mx/Luc reporter gene [16] were propagated in EMEM supplemented with 10 mM BES and 7% HS.

The pGPK cells were isolated from kidney tissue of specific-pathogen-free (SPF) guinea pigs (Slc:Hartley; Japan SLC, Shizuoka, Japan) by treating them with approximately 400 protease units/mL of dispase II (FUJIFILM, Tokyo, Japan) in EMEM and slowly stirring at 4 °C. After the treatment, the cells were seeded in the culture flasks in EMEM supplemented with 0.295% TPB and 10% BVDV-free fetal bovine serum (Japan Bio Serum, Hiroshima, Japan). The cells were incubated at 37 °C in the presence of 5% CO_2_.

A live attenuated vaccine-derived CSFV clone, vGPE^−^, was derived from pGPE^−^ [9], a plasmid containing its full-length cDNA. All mutant CSFVs, vGPE^−^/A3572P, vGPE^−^/G330R; A374D; K761R; R1134K; N1333T; A2563V; I2791L; K2792T (vGPE^−^/MS8), and vGPE^−^/G330R; A374D; K761R; R1134K; N1333T; A2563V; I2791L; K2792T; A3572P (vGPE^−^/MS9) were similarly derived from their corresponding plasmids: pGPE^−^/A3572P, pGPE^−^/G330R; A374D; K761R; R1134K; N1333T; A2563V; I2791L; K2792T (pGPE^−^/MS8), and pGPE^−^/G330R; A374D; K761R; R1134K; N1333T; A2563V; I2791L; K2792T; A3572P (pGPE^−^/MS9), respectively.

### 2.2. Plasmid Constructs

A full-length cDNA plasmid encoding the master seed GPE^−^-identical amino acid sequence, pGPE^−^-master seed (pGPE^−^-MS) was constructed in the pGPE^−^ backbone [9] using site-directed mutagenesis. Mutagenesis was conducted using the KOD FX Neo (TOYOBO, Osaka, Japan) and the Phusion High-Fidelity DNA Polymerase (New England Biolabs, Ipswich, MA, USA), along with specific primers and the In-Fusion HD Cloning Kit (TaKaRa Bio, Shiga, Japan). Mutant full-length cDNA plasmids, pGPE^−^/S3571R, pGPE^−^/A3572P, and pGPE^−^/S3571R; A3572P, were constructed in the pGPE^−^ backbone using the Phusion High-Fidelity DNA Polymerase and the In-Fusion HD Cloning Kit mentioned above. The pGPE^−^/MS8 and pGPE^−^/MS9 plasmids were similarly constructed in the pGPE^−^-MS backbone.

Three replicon cDNA plasmids that possess all possible combinations of the amino acid substitutions in NS5B (pGPE^−^-N^pro^-Luc-IRES-NS3/S3571R, pGPE^−^-N^pro^-Luc-IRES-NS3/A3572P, and pGPE^−^-N^pro^-Luc-IRES-NS3/S3571R; A3572P) were constructed in the pGPE^−^-N^pro^-Luc-IRES-NS3 backbone [9], following the same mutagenesis procedures as above. The pGPE^−^-N^pro^-Luc-IRES-NS3/GAA plasmid of a replication-deficient replicon was previously described [8]. Nucleotide sequences of the plasmids used in this study were confirmed using the ABI 3500 Genetic Analyzer (Thermo Fisher Scientific).

### 2.3. Rescue of vGPE^−^ and Mutant Viruses

The cDNA-derived viruses were rescued, as described previously [9,17]. Briefly, the plasmid was linearized at the SrfI site located at the 3′ end of the viral genomic cDNA sequence and purified. The linearized product was used for run-off transcription with the MEGAscript T7 kit (Thermo Fisher Scientific). After DNase I digestion and purification on MicroSpin S-400 HR columns (GE Healthcare, Chicago, IL, USA), RNA was transfected into SK-L cells by electroporation using a Gene Pulser Xcell Electroporation Systems (Bio-Rad, Hercules, CA, USA), set at 200 V and 500 µF, followed by incubation at 37 °C for three days. The virus recovery was confirmed by immunoperoxidase staining using an anti-NS3 monoclonal antibody (mAb) 46/1, as described previously [18]. The entire genomes of the rescued viruses were verified by sequencing using the ABI 3500 Genetic Analyzer.

### 2.4. Luciferase Activity of Viral RNA Replicase Complex

Replicon RNA was transcribed in vitro from linearized plasmid DNA with SrfI, as described above. In order to assess the efficiency of viral genome replication by luciferase expression, 6.3 × 10^6^ CPK cells were electroporated using 1 µg of replicon RNA. Electroporation was conducted at 200 V and 500 µF, as described above. The cells were incubated at 37 °C in the presence of 5% CO_2_. After 3, 12, 24, and 48 h of transfection, cell extracts were prepared with a passive lysis buffer. Firefly luciferase activities were measured using the Dual-Luciferase Reporter Assay System and the GloMAX Discover System (Promega, Madison, WI, USA). The luciferase activity was normalized with that of the replication-deficient rGPE^−^-N^pro^-Luc-IRES-NS3/GAA replicon at the initial time point.

### 2.5. In Silico Mutagenesis

The dStability values, showing the relative thermostability of the mutations as regards the vGPE^−^ type (S3571 and A3572), were calculated by using the Residue Scan tool in the Molecular Operating Environment (MOE) software version 2018 (Chemical Computing Group, Montreal, QC, Canada).

### 2.6. Virus Titration

The virus titers were determined using end-point dilution with SK-L cells. After four days of incubation at 37 °C in the presence of 5% CO_2,_ viral NS3 was detected by immunoperoxidase staining with mAb, 46/1. The titers were calculated and expressed as 50% tissue culture infective dose (TCID_50_) per mL [19].

### 2.7. Virus Growth Kinetics

The growth kinetics of the vGPE^−^ virus and its mutant viruses in the SK-L or pGPK cells were determined by inoculating them into confluent cell monolayers at a multiplicity of infection (MOI) of 0.001. After inoculation, the SK-L cells were incubated at 40 °C in the presence of 5% CO_2_ and the pGPK cells at 30 °C in the presence of 5% CO_2_. The supernatants were collected on 0, 1, 2, 3, 4, 5, 6, and 7 days post-infection (dpi), and the virus titers were measured in SK-L cells, as described above.

### 2.8. IFN Bioassay

The bioactivity of swine IFN-α/β was assessed as described previously [7]. Briefly, the supernatants of SK-L cells inoculated with viruses were inactivated using a UV cross-linker (DNA-FIX DF-254; ATTO, Tokyo, Japan) and added to the SK6-MxLuc cells. Recombinant swine IFN-α [10] served as the standard. Cell extracts were prepared with 100 µL passive lysis buffer, and the firefly luciferase activities were measured using the Dual-Luciferase Reporter Assay System and POWERSCAN 4 (DS Pharma Biomedical, Osaka, Japan). Results were recorded for three independent experiments, and each experiment was conducted in duplicate.

### 2.9. Experimental Infection of Pigs

To assess the pathogenicity of vGPE^−^, vGPE^−^/MS8, and vGPE^−^/MS9 viruses in pigs, five 2-week-old crossbred Landrace×Duroc×Yorkshire SPF pigs (Yamanaka Chikusan, Hokkaido, Japan) per group were inoculated intramuscularly with 10^7.0^ TCID_50_ of virus from the supernatant of the cell culture. As described previously, according to a defined scoring system, the pigs were monitored daily for body temperature and clinical scores for 14 days [20]. Their blood was collected in tubes containing EDTA (Venoject II VP-NA050K; Terumo, Tokyo, Japan) on 0, 3, 5, 7, 9, 11, and 14 dpi. Their serum was collected in tubes (Venoject II VP-P075K; Terumo) on 0 and 14 dpi. The serum neutralization test was used to confirm the absence of neutralizing antibodies against CSFV in the naïve pigs. The total numbers of leukocytes and platelets were counted with a pocH-100iV Diff apparatus (Sysmex, Hyogo, Japan). All pigs were euthanized on 14 dpi, and their tissues from tonsils, spleens, and mesenteric lymph nodes were aseptically collected. The tissue samples were homogenized in EMEM to obtain 10% suspension used for virus titration. The virus titers were expressed as TCID_50_ per ml (blood) or gram (tissue).

### 2.10. Serum Neutralization Test (SNT)

As previously described, luciferase-based SNT was conducted [21]. Each equal volume of serum and 100 TCID_50_ of vCSFV-GPE^−^/HiBiT [22] were mixed and incubated at37 °C for one hour. The mixture and SK-L cell suspension were incubated in 96-well plates at 37 °C and 5% CO_2_. The neutralizing antibody titers were determined on 4 days post-inoculation by the luciferase assay using the Nano-Glo HiBiT Lytic Detection System (Promega) according to the manufacturer’s instructions. The luciferase activity was measured using POWERSCAN 4 (DS Pharma Biomedical).

### 2.11. Ethics Statement

The animal experiments were authorized by the Institutional Animal Care and Use Committee of the Faculty of Veterinary Medicine, Hokkaido University (approval numbers 18-0038, approved on 26 March 2018 and 21-0023, approved on 8 April 2021), and performed according to the guidelines of this committee. The facilities where the animal experiments were conducted are certified by the Association for Assessment and Accreditation of Laboratory Animal Care International (AAALAC International).

## 3. Results

### 3.1. Loss of the Infectious Master Seed GPE^−^ Clone Production

The 10 amino acid substitutions between the master seed GPE^−^ and vGPE^−^ reside at positions 330 and 374 in E^rns^, 761 in E2, 1134 and 1333 in NS2, 2563 in NS4B, 2791 and 2792 in NS5A, and 3571 and 3572 in NS5B (Figure 1). The 10 amino acid substitutions were introduced in the vGPE^−^ backbone using multiple site-directed mutageneses to generate the infectious cDNA clone with identical amino acid sequence to the master seed GPE^−^ (vGPE^−^-MS) (Figure 2a). Subsequently, the full-length viral RNA of vGPE^−^-MS was electroporated into the swine kidney-derived SK-L cells. However, the viral NS3 protein was not detected. This indicates the absence of the infectious vGPE^−^-MS virus (Figure 2c).

### 3.2. Critical Amino Acid Residue for Virus Production

Since the absence of vGPE^−^-MS must be associated with one or some of the 10 introduced substitutions, the amino acid sequence of the master seed GPE^−^ was compared with those of several representative CSFV strains (Figure 3). According to the alignment, T1333 (in NS2), T2792 (in NS5A), R3571, and P3572 (in NS5B) were likely to be the amino acids unique to the master seed GPE^−^. Notably, R3571 and P3572 reside in NS5B, a viral RNA-dependent RNA polymerase (RdRp). As the viral polymerase is well conserved in the virus species, even in the genus or family, its alteration can directly result in viral replication deficiency. The two consecutive residues were targeted as a probable cause of the loss of infectious vGPE^−^-MS production. To this end, S3571R and A3572P substitutions were introduced separately or in combination in the vGPE^−^ backbone, and in vitro-transcribed RNAs of vGPE^−^/S3571R, vGPE^−^/A3572P, and vGPE^−^/S3571R; A3572P were subjected to electroporation (Figure 2b). The detection of NS3 confirmed virus production in the same way as above. As expected, vGPE^−^/S3571R; A3572P, a virus that possesses the identical NS5B sequence to vGPE^−^-MS, was not rescued, and neither was vGPE^−^/S3571R rescued, while the other single amino acid-substituted virus, vGPE^−^/A3572P, was rescued (Figure 2d). Therefore, the virus production of vGPE^−^-MS was eliminated by S3571R substitution. The blind passaging of the supernatant of vGPE^−^-MS RNA-electroporated cells yielded a virus that has a glycine substitution at 3571 (data not shown). This result further supports the absence of the virus possessing arginine at 3571.

### 3.3. Impaired Viral Replication by the Amino Acid Substitution in NS5B

To further investigate the effect of substitutions, especially on the viral genome replication, the vGPE^−^-derived bi-cistronic replicon, rGPE^−^-N^pro^-Luc-IRES-NS3 was used. Three types of amino acid-substituted replicons (rGPE^−^-N^pro^-Luc-IRES-NS3/S3571R, rGPE^−^-N^pro^-Luc-IRES-NS3/A3572P, and rGPE^−^-N^pro^-Luc-IRES-NS3/S3571R; A3572P) were prepared to evaluate the replication efficiency in CPK cells based on their encoding firefly luciferase activity (Figure 4a). The replicons harboring S3571R substitution decreased luciferase activity to the same level as the replication-deficient rGPE^−^-N^pro^-Luc-IRES-NS3/GAA replicon at 24 and 48 h after electroporation (Figure 4b). Thus, the loss of S3571R-substituted virus production was attributed to the reduction of replication efficiency by S3571R substitution. Additionally, the relative thermostability of NS5B was simulated to consider the possibility of conformational disorder resulting from amino acid substitutions (Table 1). Surprisingly, it was not the replication-affecting S3571R substitution but the A3572P substitution that exhibited increased internal energy, which is likely to make the protein structure unstable.

### 3.4. In Vitro Growth Kinetics of Master Seed GPE^−^ Variants

Since S3571R and A3572P substitutions confer functional and conformational disorder to the virus, the credibility of the consecutive amino acids of the master seed GPE^−^ is debated. Therefore, the functionally fatal S3571R substitution was excluded to generate two types of replication-competent vGPE^−^-MS alternatives, vGPE^−^/MS8 and vGPE^−^/MS9 (Figure 5a,b). While vGPE^−^/MS9 possesses the identical amino acid sequence, except at 3571 to the master seed GPE^−^, the conformationally disadvantageous A3572P substitution was also excluded from vGPE^−^/MS8—a total of 8 amino acid substitutions from vGPE^−^. As the GPE^−^ strain possesses three distinct characteristics (END^–^, efficient propagation at 30 °C, and efficient propagation in guinea pig kidney cells), the growth characteristics of the seed variants in pGPK cells at 30 °C were next investigated. According to the results, vGPE^−^/MS8 and vGPE^−^/MS9 propagated to almost the same extent as vGPE^−^ at 5 dpi, albeit slower than vGPE^−^ (Figure 5c). Subsequently, the last END^–^ characteristic of the two seed variants was assessed by measuring the IFN-α/β production in virus-infected SK-L cells. As previously reported, vGPE^−^ was incapable of suppressing IFN production and induced a large amount of IFN-α/β in the infected cells [7]. In contrast, a slight amount of IFN-α/β was induced by vGPE^−^/MS8 and vGPE^−^/MS9. In the same infected cells, vGPE^−^/MS8 and vGPE^−^/MS9 propagated more efficiently than vGPE^−^ at 1 dpi, but their growth was gradually suppressed (Figure 5d), which was remarkable for vGPE^−^/MS9.

### 3.5. Pathogenicity of Master Seed GPE^−^ Variants in Pigs

To assess the pathogenicity of vGPE^−^/MS8 and vGPE^−^/MS9 in pigs, three groups of five pigs were inoculated intramuscularly with 10^7.0^ TCID_50_ of vGPE^−^, vGPE^−^/MS8, and vGPE^−^/MS9, respectively. Distinct elevation in the body temperature was not recorded in 14 days of the experiments. Likewise, almost all pigs developed no clinical signs throughout the experiments (Appendix A). One vGPE^−^/MS9-infected pig, exceptionally, showed a mild diarrhea sign only on 12 dpi without further progression, which is assumed to be a disease-unrelated clinical sign. Neither typical leukopenia nor thrombocytopenia was observed in any pigs investigated. The number of leukocytes varied in each group at the late stage of infection (Appendix A). According to the previous studies [7,9], a few vGPE^−^-infected pigs exhibited signs of transient viremia and very low titers of viruses were recovered from their blood for 14 days after infection. The vGPE^−^/MS9-infected pigs also showed transient viremia, but the period of viremia of vGPE^−^/MS8-infected pigs was prolonged (Table 2). On 14 dpi, viruses were recovered from the tonsils of pigs in all groups, which are the primary site of their infection. No virus was recovered from the collected organs except for the mesenteric lymph node of a vGPE^−^/MS9-infected pig. SNT was used to measure neutralizing antibodies in the sera collected on 14 dpi. Notably, vGPE^−^ induced the neutralizing antibodies in all five pigs on 14 dpi, and the antibody titers were higher than those of vGPE^−^/MS8- or vGPE^−^/MS9-infected pigs (Table 2).

## 4. Discussion

Here, the generation of the original GPE^−^ vaccine seed clone, vGPE^−^-MS, was attempted with the reverse genetics system. However, the serine-to-arginine substitution at 3571 in the viral NS5B polymerase abrogated viral replication, resulting in vGPE^−^-MS production loss. Thus, two seed variants, vGPE^−^/MS8 and vGPE^−^/MS9, were generated. Regarding the sequence homology, vGPE^−^/MS9 is the most identical strain to the master seed GPE^−^. However, the proline sequence at 3572 is unreliable to the same extent as the arginine sequence at 3571, considering its consequent structural disorder. Perhaps the substitutions at 3571 and 3572 can be explained by flipping the third and fourth nucleotides in the sequential codons encoding these two amino acids (AGC GCC to AGG CCC). In any case, it is quite difficult to exclude the possibility of a human error in the NS5B sequence of the master seed GPE^−^. Therefore, two seed variants, vGPE^−^/MS8 and vGPE^−^/MS9, should be appreciated as subjects to characterize the GPE^−^ vaccine seed.

In this study, both vGPE^−^/MS8 and vGPE^−^/MS9, and vGPE^−^ met vaccine-specific characteristics: propagation in pGPK cells at 30 °C and slight propagation in vivo. However, their characteristics differed, and the difference was obvious, especially in the capacity of IFN-α/β production in swine cells. The inhibition of type I IFN production in infected cells is one of the essential pestivirus functions in evading the host’s innate immune system [1]. This inhibitory function becomes apparent as END phenomenon in the cells superinfected with the Newcastle disease virus, and the END phenomenon was widely used for detecting noncytopathogenic CSFV or BVDV in their infected cells [11,23,24,25]. Regarding the GPE^−^ strain, it cannot suppress type I IFN production in its infected cells and is described as an END^–^ strain [2,13,26]. Previous studies have highlighted several amino acids in the TRASH zinc-binding domain of N^pro^ as the determinants in suppressing IFN production through IRF3 degradation [12,27]. Especially, the amino acid at the position of 136 in the zinc-binding domain is responsible for the END^–^ phenotype of the GPE^−^ strain, and the N136D substitution in the vGPE^−^ backbone contributes to the enhancement of pathogenicity in pigs [7]. This study showed the significantly reduced IFN-α/β induction by vGPE^−^/MS8 and vGPE^−^/MS9 in SK-L cells, as previously shown by the END phenomenon positive (END^+^) GPE^−^ strain, vGPE^−^/N136D [7]. Given that both seed variants lack the N^pro^-mediated IRF3 degradation, one of the responsible factors is probably the viral envelope protein E^rns^. The E^rns^ is a pestivirus unique secreted protein, which suppresses type I IFN induction by degrading single-stranded and double-stranded RNAs with its RNase activity [28,29,30,31]. Despite the abrogation of the viral RNase activity affects the cytopathogenicity and pathogenicity [32,33], amino acid substitutions in the E^rns^ of vGPE^−^/MS8 and vGPE^−^/MS9 reside outside of the RNase domain. In a previous report, the replacement of the C-terminal of the core protein and the full-length of E^rns^ of a CSFV vaccine strain resulted in the conversion of the END phenotype without the abrogation of its RNase activity [5]. Combined with this, this study might propose alternative critical residues for regulating the type I interferon induction, as in the case of N^pro^ [34]. However, synergetic effects with other amino acid substitutions should be considered to clarify the underlying mechanisms.

Overall, the propagation of vGPE^−^/MS9 was relatively lower, compared to the other two strains, in vitro and in vivo. The in silico study suggested that the alanine-to-proline substitution at the position of 3572, right behind 3571, was likely to make the conformation of the NS5B unstable, whereas the fatally affecting S3571R substitution was not. CSFV NS5B consists of four domains: the N-terminal domain (NTD), the finger domain, the palm domain, and the thumb domain [35,36]. The amino acid at 3571 resides in the middle finger of the finger domain but is far from the palm domain that contains the catalytic core of the RdRp (Appendix A). Hence, it is reasonable to speculate that the impaired replication was not caused by the disruption of the polymerase activity but by some yet unclarified functions of NS5B, such as the intramolecular interactions with the pestivirus unique NTD. As previously reported, NTD affected the fidelity of NS5B by interacting with the palm domain [36] and not with the finger domain. However, note that the conformation of NTD might vary in viruses, as that of the BVDV NADL strain was probably different from CSFV NS5B [35,37]. A notable study on the N-terminal methyltransferase (MTase) was recently reported in the genus of *Flavivirus* [38]. The flaviviral polymerase NS5 harbors an MTase, as the pestivirus NS5B harbors an NTD, in two different conformations (Appendix A). The study suggested that the flaviviral NS5 would undergo a conformational change from one to the other form during the replication. Significantly, one of the key residues for the MTase-RdRp interaction resides at a position similar to the CSFV 3571 residue (Appendix A). As the RdRp cores of CSFV NS5B (115–678) and Japanese encephalitis virus (JEV) NS5 (304–895) share a relatively high similarity with the root mean square deviation (RMSD) of 2.846, further study on the pestivirus NTD would be essential to unravel this mechanism. The virus possessing glycine at 3571, which was produced after blind passaging, also supports the hypothesis that the substitution at 3571 affects some replication-modulating interactions but not its polymerase activity.

The experimental conditions taken in this study do not conform to the vaccination instructions and ideal observation period, i.e., vaccination of 30–40-day-old pigs and monitoring for three to four months, which give the pigs enough antibody response [13]. Nevertheless, the difference in the END^–^ phenotype between vGPE^−^ and seed variants should be considered to propose a preferable backbone for a new marker vaccine. Generally, the END^–^ phenotype, mirrored by the high capacity of type I interferon induction, is an important property that ensures safety and effectiveness. It enables the vaccine to induce a strong host immune response without CSF-typical immunosuppression [39]. Moreover, that distinct phenotype is recognized as a marker to keep the authenticity of vaccine products. In this study, some in vivo virus characteristics suggested the correlations with the capacity of IFN-α/β induction. The highly IFN-α/β-inducing vGPE^−^-infected pigs showed transient viremia and high antibody response with the geometric mean-neutralizing antibody titer of 4. In contrast, the poorly IFN-α/β-inducing vGPE^−^/MS8-infected pigs showed prolonged viremia and low antibody response with the geometric mean-neutralizing antibody titer of less than 2. Although the in vivo characteristics of the vGPE^−^/MS9 did not correlate with its IFN-α/β induction as the other two strains did, it is reasonable to believe that was provided by the protein stability-affecting A3572P substitution. Following these results, the distinct vaccine-specific characteristics of vGPE^−^ must be preferable in terms of safety and effectiveness. In addition, the reverse genetics system-based production of vGPE^−^ will stably ensure the vaccine quality. These advantages of vGPE^−^ would position it as an ideal backbone for the future development of a CSF marker vaccine. However, again, the limitation of the present experimental conditions has to be substantially considered to make a final conclusion. Especially, the safety of the pregnant sows needs to be carefully investigated to prevent the transplacental transmission of the virus.

Though how vGPE^−^ acquired the eight or more amino acid substitutions is undefined, it is possible that the reference strain of vGPE^−^ already possessed some of the substitutions before the passages in FS-L3 cells. Given the long attenuation process of the GPE^−^ strain, the reference strain of vGPE^−^ might also be a variant of the vaccine seed, which emerged in the attenuation process. Although it can no longer identify the exact characteristics of the GPE^−^ vaccine products, which greatly contributed to the past CSF eradication, the newly discovered characteristics of GPE^−^ variants indicate a way for CSF re-eradication. Further investigations on the effects of each amino acid substitution would support the elucidation of the determinants of GPE^−^ attenuation and CSFV pathogenesis.

## Figures and Tables

**Figure 1 viruses-13-01672-f001:**
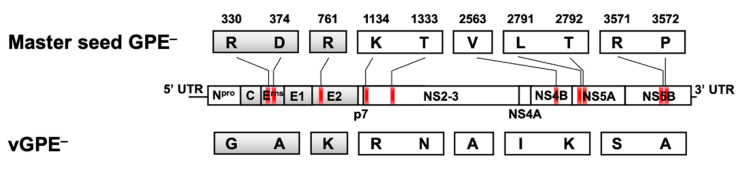
The difference in amino acid sequence between the master seed GPE^−^ and vGPE^−^. The coding regions of the master seed GPE^−^ (GenBank Accession Number D49533.1) and vGPE^−^ are displayed as boxes divided by each protein gene. Structural proteins and nonstructural proteins are shown in gray and in white, respectively. A total of 10 amino acid substitutions are recognized at positions 330 and 374 in E^rns^, 761 in E2, 1134 and 1333 in NS2, 2563 in NS4B, 2791 and 2792 in NS5A, and 3571 and 3572 in NS5B. The substitution positions are shown in red on the backbone of vGPE^−^.

**Figure 2 viruses-13-01672-f002:**
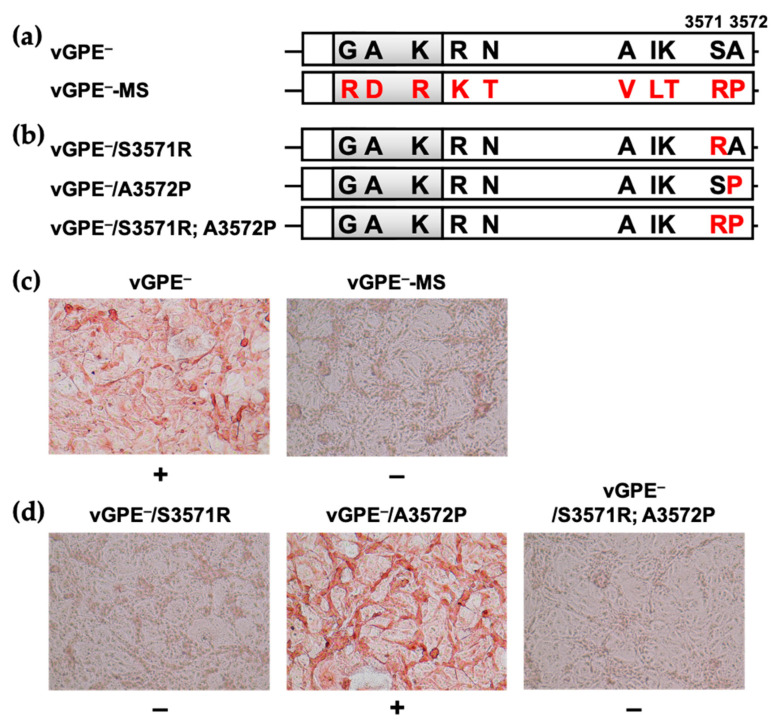
SK-L cells electroporated with in vitro-transcribed RNAs. The amino acid substitutions (red) were introduced in the backbone of vGPE^−^. (**a**) the viral genome of vGPE^−^-MS possesses the identical amino acid sequence to the master seed GPE^−^; (**b**) three viral genomes of vGPE^−^/S3571R, vGPE^−^/A3572P, vGPE^−^/S3571R; A3572P possess substitutions in NS5B; (**c**,**d**) the in vitro-transcribed RNAs derived from the viral genomes above were electroporated into SK-L cells. After 72 h, the cells were heat-fixed and immunostained using the anti-NS3 mAb, 46/1. + depicts that the antibody reacted with the viral NS3 proteins: the success of virus rescue.—depicts that the antibody did not react with the viral NS3: failure of virus rescue. The original magnification was ×200.

**Figure 3 viruses-13-01672-f003:**
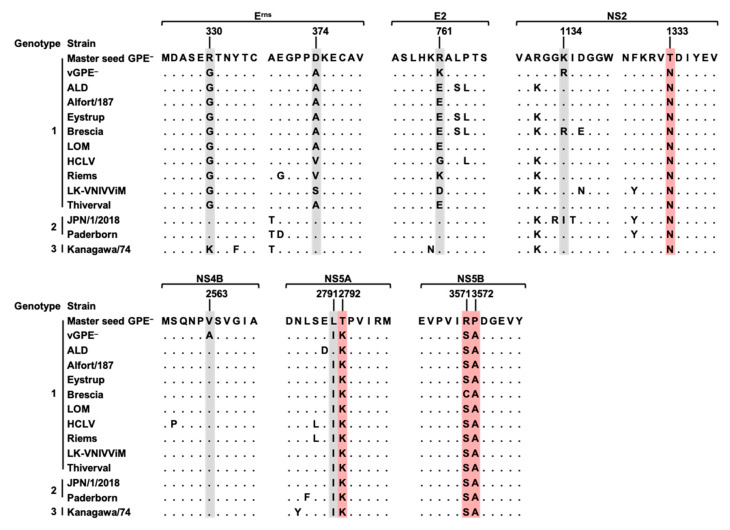
Amino acid alignment of substitution sites among CSFV strains. The amino acid sequences of 10 substitution sites and their contexts of selected CSFV strains are aligned. The sites of amino acid substitution are highlighted in gray. The amino acid sequences unique to the master seed GPE^−^ are highlighted in red.

**Figure 4 viruses-13-01672-f004:**
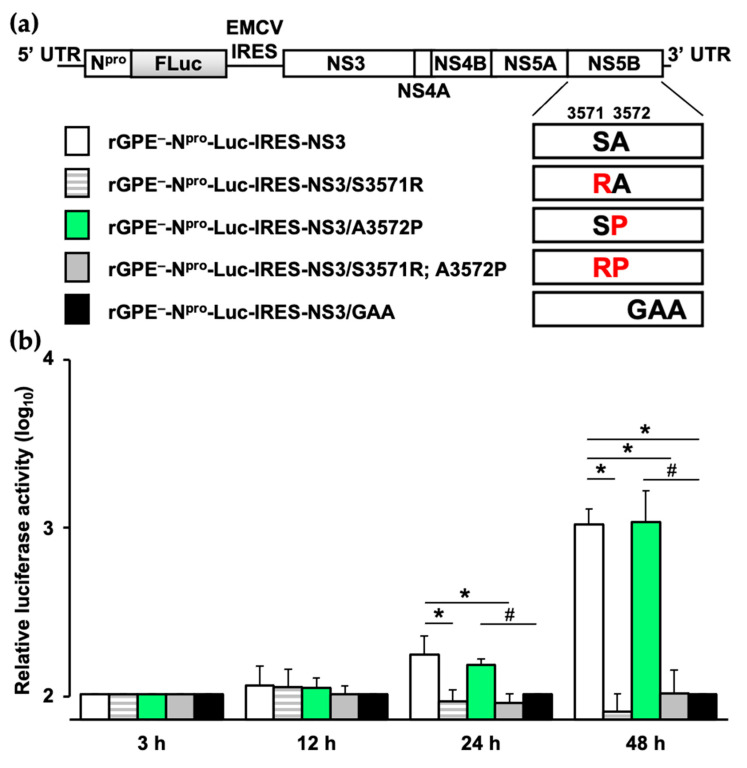
Effect of amino acid substitutions in NS5B on the viral RNA replication efficiency: (**a**) schematic diagrams of replicon genomes, coding firefly luciferase. The substituted amino acids are shown in red at positions 3571 and 3572 of NS5B; (**b**) the in vitro-transcribed replicon RNAs were electroporated into CPK cells. After 3 h, 12 h, 24 h, and 48 h of incubation, the firefly luciferase activity, expressed in relative light units, was assayed in cell lysates. The firefly luciferase activity was normalized with that of the replication-deficient rGPE^−^-N^pro^-Luc-IRES-NS3/GAA replicon at the initial time point. Each bar represents the mean value of the triplicates, with error bars showing the standard deviations. The significance of relative luciferase activity differences with replication-competent rGPE^−^-N^pro^-Luc-IRES-NS3 or replication-incompetent rGPE^−^-N^pro^-Luc-IRES-NS3/GAA was calculated with Student’s *t*-test. * indicates *p* < 0.05 compared with the rGPE^−^-N^pro^-Luc-IRES-NS3. # indicates *p* < 0.05 compared with the rGPE^−^-N^pro^-Luc-IRES-NS3/GAA.

**Figure 5 viruses-13-01672-f005:**
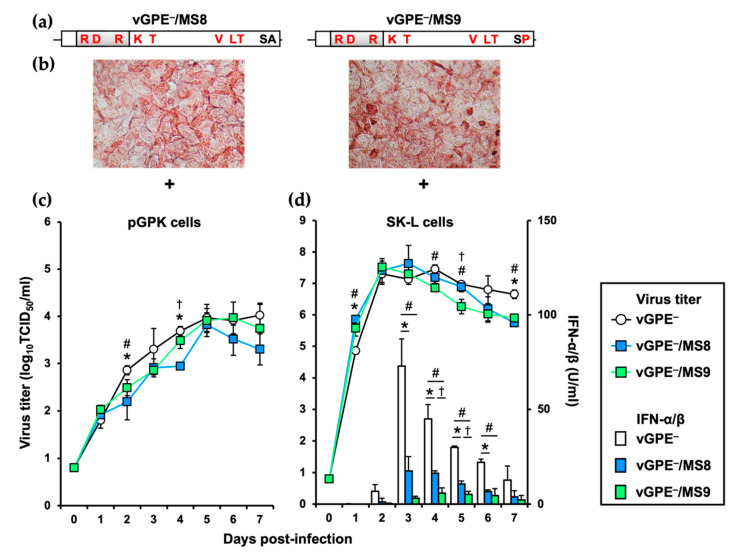
Characteristics of vGPE^−^, vGPE^−^/MS8, and vGPE^−^/MS9 in vitro: (**a**) the viral genomes of vGPE^−^/MS8 and vGPE^−^/MS9 are shown; (**b**) the in vitro-transcribed RNAs of vGPE^−^/MS8 and vGPE^−^/MS9 were electroporated into the SK-L cells. After 72 h, the cells were heat-fixed and immunostained using the anti-NS3 mAb, 46/1. + depicts that the antibody reacted with the viral NS3 proteins: the success of virus rescue.—depicts that the antibody did not react with the viral NS3: failure of virus rescue. The original magnification was ×200; (**c**,**d**) the pGPK cells and SK-L cells were infected with vGPE^−^, vGPE^−^/MS8, or vGPE^−^/MS9 at a multiplicity of infection (MOI) of 0.001. The pGPK and SK-L cells were incubated at 30 °C and 40 °C in the presence of 5% CO_2_, respectively. The supernatants were collected at indicated times. The titers were measured in SK-L cells and expressed as 50% tissue culture infective dose (TCID_50_) per mL. The IFN-α/β bioactivity in the supernatants of SK-L cells was measured using the SK6-MxLuc cells. Error bars represent the standard deviations. The significance of mutant viral growth differences was calculated using one-way ANOVA followed by Student’s *t*-test with Bonferroni correction. * indicates *p* < 0.025 between vGPE^−^ and vGPE^−^/MS8. # indicates *p* < 0.025 between vGPE^−^ and vGPE^−^/MS9. † indicates *p* < 0.025 between vGPE^−^/MS8 and vGPE^−^/MS9.

**Table 1 viruses-13-01672-t001:** Change of stability by amino acid substitutions with in silico mutagenesis calculation.

Amino Acid (AA)		Corresponding Virus with the AAs	dStability (kcal/mol)
3571	3572		
S	A	vGPE^−^, vGPE^−^/MS8	0.000000
R	A	vGPE^−^/S3571R	0.539012
S	P	vGPE^−^/A3572P, vGPE^−^/MS9	1.265029
R	P	vGPE^−^-MS	1.640682

**Table 2 viruses-13-01672-t002:** Virus recovery from and seroconversion in pigs infected with vGPE^−^ and seed variants.

Virus	Virus Recovery from:										Seroconversion	
	Blood (log_10_ TCID_50_/mL) ^a^ on dpi:							Tissue (log_10_ TCID_50_/g) ^b^			Neutralizing Antibody Titer on dpi:	
	0	3	5	7	9	11	14	Tonsil	Spleen	Mesenteric Lymph Node	0	14
vGPE^−^	–	–	≤1.0	–	–	–	–	+	–	–	<1	1
	–	–	–	–	–	–	–	+	–	–	<1	4
	–	–	–	–	–	–	–	≤2.0	–	–	<1	2
	–	–	–	–	–	–	–	–	–	–	<1	16
	–	+	+	–	–	–	–	≤2.0	–	–	<1	8
vGPE^−^/MS8	–	–	+	–	–	–	–	+	–	–	<1	2
	–	–	+	–	–	–	–	≤2.1	–	–	<1	1
	–	–	+	+	+	–	–	≤2.3	–	–	<1	<1
	–	–	–	–	+	–	–	+	–	–	<1	8
	–	–	+	–	+	–	–	≤2.7	–	–	<1	1
vGPE^−^/MS9	–	–	–	–	–	–	–	+	–	–	<1	<1
	–	–	–	+	–	–	–	≤2.5	–	–	<1	<1
	–	–	–	–	–	–	–	≤2.0	–	–	<1	<1
	–	–	–	–	–	–	–	+	–	–	<1	1
	–	–	+	–	–	–	–	≤2.5	–	+	<1	<1

^a^–: not isolated, +: isolated in a 6-well plate and was lower than the detection limit of TCID_50_ (10^0.8^ TCID_50_/mL) in a 96-well plate. ^b^–: not isolated, +: isolated in a 6-well plate and was lower than the detection limit of TCID_50_ (10^1.8^ TCID_50_/g) in a 96-well plate.

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
