# Peer review of "Characteristics of Classical Swine Fever Virus Variants Derived from Live Attenuated GPE Vaccine Seed"

_viruses, 2021, doi:10.3390/v13081672_

Round 1

Reviewer 1 Report

A huge amount of work has been done and provided data has a big interest to the scientific community, especially after many years of attention, mainly to african swine fever vaccine development. 

But the suggestion of "the vGPE–  mainly  retains ideal properties for the CSF  vaccine,  compared  with  the  seed  variants, and  is  probably  useful  in the development of  a CSF marker vaccine" need to be clarifide from the practical point of view. 

Live attenuated vaccine strains  show a very limited replication in target animals. Therefore, from this point using PCR protocol for DIVA purpose in field conditions are not useful in many cases.

Suppose authors plan to use the serological marker system of "vGPE–”. In that case, testing schemes should take the obvious limitations in DIVA diagnostics into consideration and the latter should be reflected in the sample size. It should also be discussed to put more emphasis on
the detection of the field virus itself and genetic DIVA.

Reviewer 2 Report

Abstract – The paper represents some very good work and characterization of the virus. But I am not sure why the work was undertaken. A bit more information regarding the goals and purposes as well as some justification would be useful to the reader.

Line 53 – suggested: Some vaccines are referred to as as differentiating infected from vaccinated animals (DIVA) vaccines.

The wording of this paragraph (beginning at line 53) is confusing and needs clarification – especially the second and third sentences.

Line 63 - The attenuation of the GPE– strain is represented by results in restricted propagation in pigs.

Line 65 – minimal isolation or minimal shedding? Slight virus recovery implies a clinical disease recovery.

The sentence at line 67 does not make sense – “exaltation”?

Line 77 – the paragraph is overly wordy and it confuses the reader.

The materials and methods are adequate and provide sufficient detail with references for understanding and interpretation of the experimental work.

Line 273 – Is “energy” the intended word?

Line 324 – Pathogenicity….

                I see some relative virulence and infectivity data. This is a good study. I would suggest it be titled – Relative Virulence and Infectivity. Why was there no substantially virulent control used? There was no evaluation of actual pathology or pathogenesis.

Discussion – This is a good and thorough discussion.
